# Characterization of the Fermentation and Sensory Profiles of Novel Yeast-Fermented Acid Whey Beverages

**DOI:** 10.3390/foods10061204

**Published:** 2021-05-27

**Authors:** Siyi (Rossie) Luo, Timothy A. DeMarsh, Dana deRiancho, Alina Stelick, Samuel D. Alcaine

**Affiliations:** Department of Food Science, Cornell University, Ithaca, NY 14853, USA; sl3288@cornell.edu (S.L.); tad24@cornell.edu (T.A.D.); dld238@cornell.edu (D.d.); ap262@cornell.edu (A.S.)

**Keywords:** acid whey, dairy, sustainability, fermented beverage, waste products

## Abstract

Acid whey is a by-product generated in large quantities during dairy processing, and is characterized by its low pH and high chemical oxygen demand. Due to a lack of reliable disposal pathways, acid whey currently presents a major sustainability challenge to the dairy industry. The study presented in this paper proposes a solution to this issue by transforming yogurt acid whey (YAW) into potentially palatable and marketable beverages through yeast fermentation. In this study, five prototypes were developed and fermented by *Kluyveromyces marxianus*, *Brettanomyces bruxellensis*, *Brettanomyces claussenii*, *Saccharomyces cerevisiae* (strain: Hornindal kveik), and IOC Be Fruits (IOCBF) *S. cerevisiae*, respectively. Their fermentation profiles were characterized by changes in density, pH, cell count, and concentrations of ethanol and organic acids. The prototypes were also evaluated on 26 sensory attributes, which were generated through a training session with 14 participants. While *S. cerevisiae* (IOCBF) underwent the fastest fermentation (8 days) and *B. claussenii* the slowest (21 days), *K. marxianus* and *S. cerevisiae* (Hornindal kveik) showed similar fermentation rates, finishing on day 20. The change in pH of the fermentate was similar for all five strains (from around 4.45 to between 4.25 and 4.31). Cell counts remained stable throughout the fermentation for all five strains (at around 6 log colony-forming units (CFU)/mL) except in the case of *S. cerevisiae* (Hornindal kveik), which ultimately decreased by 1.63 log CFU/mL. *B. bruxellensis* was the only strain unable to utilize all of the sugars in the substrate, with residual galactose remaining after fermentation. While both *S. cerevisiae* (IOCBF)- and *B. claussenii*-fermented samples were characterized by a fruity apple aroma, the former also had an aroma characteristic of lactic acid, dairy products, bakeries and yeast. A chemical odor characteristic of petroleum, gasoline or solvents, was perceived in samples fermented by *B. bruxellensis* and *K. marxianus*. An aroma of poorly aged or rancid cheese or milk also resulted from *B. bruxellensis* fermentation. In terms of appearance and mouthfeel, the *S. cerevisiae* (IOCBF)-fermented sample was rated the cloudiest, with the heaviest body. This study provides a toolkit for product development in a potential dairy-based category of fermented alcoholic beverages, which can increase revenue for the dairy industry by upcycling the common waste product YAW.

## 1. Introduction

Acid whey, a by-product generated during the production of fresh cheese and Greek yogurt, has long presented a sustainability challenge to the dairy industry [1]. In the past two decades, there has been a significant increase in consumer interest in Greek yogurt, due to its high protein content (5.6%) and richer consistency, leading to enormous quantities of acid whey being generated as waste [2]. Unlike sweet whey, a by-product of hard cheese production which has a higher protein content (5.11 ± 0.07 g/L) and pH (6.0–6.5), acid whey currently has limited applications with regard to the manufacture of value-added products, and presents challenges during disposal. Sweet whey is often processed into whey protein powder [3], while acid whey’s only current applications include use in animal feed and as a fertilizer for agricultural crops. Approximately 300,000 tons of acid whey were generated in New York State alone in 2012 [4]. The low pH (3.5–5) and high chemical oxygen demand (52,400 to 62,400 mg/L) of acid whey makes it unsuitable for discharge directly into natural water streams or municipal waters, as there is concern of causing algae blooms and upsetting the equilibrium of ecosystems [5,6]; acid whey may be treated in wastewater treatment plants to allow for proper disposal [4]. Recent research in various disciplines has proposed solutions to tackle this problem, such as turning acid whey into a source of value-added compounds through biotechnology [7,8], and utilizing acid whey as an ingredient in food products [1]. However, acid whey still contains untapped potential to be utilized as the substrate for fermented beverages. This could introduce a new, much-needed stream of revenue for dairy producers. According to Menchik and colleagues (2019) [6], although yogurt acid whey (YAW) is low in protein (up to 3.71 mg/g), it contains up to 3.5% lactose, making it a suitable substrate for yeast fermentation. In addition, its relatively high concentrations of calcium (128 mg/100 g) and other minerals may increase the appeal of fermented YAW beverages to health-conscious consumers. 

The total US sales of alcoholic beverages reached $242.2 billion in 2018, among which fermented alcoholic beverages such as beer and alcoholic cider accounted for $114.5 billion [9,10]. With improvements in environmental education and regulations in recent decades, consumers are becoming increasingly aware of the importance of sustainability, and are showing a preference for eco-friendly products over conventional products [11,12]. Therefore, there is ample space in the alcoholic beverage market for a new product category, particularly one that promotes sustainability. 

*Saccharomyces cerevisiae* is a common yeast species with diverse applications in the food and beverage industry. It is well known for its role in the production of fermented beverages, including beer, wine, cider, and distilled beverages. In particular, it has been a topic of interest to the wine industry to understand the complex metabolic pathways exhibited by *S. cerevisiae,* and their relation to sensory characteristics of fermented products produced by the organism. The main groups of compounds produced by *S. cerevisiae* that contribute to the wine bouquet include higher alcohols, such as 1-butanol (fusel oil odor), isobutanol (alcoholic flavor), 2-phenylethanol (floral, rose notes) and isoamyl alcohol (marzipan flavors); esters, such as ethyl acetate (varnish, nail polish, fruity notes), phenylethyl acetate (rose, honey, and fruity or flowery notes), ethyl octanoate (pineapple, pear notes) and ethyl decanoate (floral notes); and acetaldehyde (fruity, green, grassy notes) [13]. Currently, there are a number of commercially available *S. cerevisiae* strains, each of which is used preferentially for specific categories of fermented beverage [14]. Although *S. cerevisiae* is incapable of fermenting lactose, it has been documented to utilize both sugar monomers resulting from the hydrolysis of lactose in the context of an acid whey milieu [15].

Other, lesser-known yeast species have been increasingly investigated for their potential use in the industrial production of both fermented beverages and biofuels in recent years. *Kluyveromyces marxianus* has been studied for its ability to metabolize lactose via the enzyme β-galactosidase, which makes it an ideal candidate to carry out the fermentation of whey [16,17]. Volatile compounds such as acetaldehyde (nutty, green apple notes), ethyl acetate (solvent, fruity notes), isoamyl and isobutyl alcohols (alcohol, banana, solvent notes), and acetic acid (vinegar notes) were detected in *K. marxianus*-fermented cheese whey spirits [17]. *Brettanomyces* yeasts, especially strains of *Brettanomyces bruxellensis,* are traditionally classified as spoilage organisms by wine makers, due to the undesirable flavors they produce. These are frequently described as being reminiscent of spices and cloves, or having phenolic/medicinal, earthy, barnyard, or horsy notes [18]. However, low levels of the compounds responsible for these flavors, such as 4-ethylphenol, 4-ethylguaiacol, and *N*-heterocyclic compounds, are considered to impart complexity to the sensory profiles of some wines. *Brettanomyces* strains are likely to be suitable for fermenting a YAW substrate, as they have a reputation for surviving in harsh conditions marked by low pH, high ethanol content, or poor nutrient availability [19]. While *B. bruxellensis* has been demonstrated to grow on glucose and galactose, the monomers that constitute lactose, this species is unable to utilize lactose directly [20,21]. *Brettanomyces claussenii,* however, was reported to be capable of directly fermenting lactose both aerobically and anaerobically, although not to the same extent as *K. marxianus* [21].

The aim of this study is to characterize the fermentation processes and sensory profiles of prototypes of novel yeast-fermented YAW beverages. This foundational knowledge provides a scalable process for upcycling YAW, and will increase the diversity of the alcoholic beverage market by introducing a dairy-based category. 

## 2. Materials and Methods

### 2.1. Fermentation Characterization

*K. marxianus* (ARGTD-0024)*, B. bruxellensis* (ARGSMK-0018), and *B. claussenii* (ARGTD-0007) were obtained from cryotubes (containing 69.95% inoculum, 30% glycerol and 0.05% tween 80) stored in the −80 °C freezer in the Alcaine Research Group (Cornell University, Food Science Department). *S. cerevisiae* (strain: Hornindal kveik) is a commercial strain purchased from Omega Yeast (St. Louis, MO, USA) and is listed by the manufacturer as strain OYL-091. This yeast strain was stored in the same condition and propagated using the same method (described in detail below) as the previously mentioned yeast strains obtained from the Alcaine Research Group. IOC Be Fruits (IOCBF) *S. cerevisiae* is a commercial strain of active dried yeast acquired from Lallemand Oenology (Petaluma, CA, USA). All of the yeast strains used in this study, with the exception of *S. cerevisiae* (IOCBF), were first streaked onto potato dextrose agar medium with 0.0025% chloramphenicol (*w*/*v*), and were incubated at 30 °C until single colonies were obtained. A broth containing 120 g of dry malt extract (DME) (Briess Malt and Ingredients Co., Chilton, WI, USA) per liter of Milli-Q water (Milli-Q Advantage A10 system, MilliporeSigma, Burlington, MA, USA) was prepared for subsequent propagation of the isolates; this broth was sterilized using a 0.45-micron filter. A single colony was selected representing each isolate, and was transferred to a tube containing 5 mL of DME broth. These tubes were incubated at 30 °C with shaking at 200 rpm, along with a negative control tube which was later used to verify that the cultures had been prepared using sterile technique. Because the five strains under investigation proliferate at different rates, the culture tubes were monitored for turbidity and removed from incubation once they appeared noticeably turbid when compared to the negative control tube. Each culture was then pitched into its own flask containing 100 mL of sterile DME broth. The flasks were incubated at 30 °C with shaking at 200 rpm until the cell number in each flask, as determined using a hemocytometer, reached a minimum of 3.20 × 10^8^ colony-forming units (CFU)/mL.

#### 2.1.1. Fermentation Setup

The YAW used in this experiment was obtained in three separate batches from a local dairy company in New York, USA, and was autoclaved upon receipt; one of the separate batches was used for each biological replicate in this experiment. Five sterile 1 L media glass bottles (Corning Inc., Corning, NY, USA) were used as fermentation vessels. Each fermentation bottle was filled with 1.03 L of sterile YAW. To the bottles that were to receive the yeast strains which are known to be unable to ferment lactose, including *B. bruxellensis*, *S. cerevisiae* (Hornindal kveik) and *S. cerevisiae* (IOCBF), 574.1 μL of DSM Maxilact A4 liquid lactase (EC 3.2.1.108) was added to 1.03 L of YAW; this volume of lactase provided an optimal enzyme to substrate ratio of 100 acid lactase units per gram of lactose, based on the assumption that there is approximately 35 g of lactose per liter of YAW [6]. For each strain to be used in the experiment, except for *S. cerevisiae* (IOCBF), a centrifuge tube was prepared containing an inoculum that would provide its fermentate with an initial cell number of 6.00 × 10^6^ CFU/mL. These inocula were centrifuged at 4000 rpm for 2 min in order to separate the cells from the DME supernatant; the supernatant was discarded, and the cells were resuspended in 10 mL of sterile YAW prior to inoculation. Separately, 618 mg of IOC Be Fruits *S. cerevisiae* (the amount needed to achieve an inoculation level of 6.00 × 10^6^ CFU/mL of substrate) was re-hydrated in 10 mL of sterile YAW for half an hour before inoculation. Following preparation of inocula, each bottle of acid whey was inoculated with its respective culture. Each bottle was then equipped with a sterile polyvinyl chloride septum that had been modified to contain a hole in its center; each bottle was also fitted with an airlock, in order to create and maintain an anaerobic environment. An InLab Smart Pro-ISM pH probe (Mettler Toledo; Columbus, OH, USA) was inserted vertically through the hole in the center of each septum, and was connected to an iCinac (AMS Alliance; Rome, Italy), providing measurements of both pH and temperature for each fermentate daily throughout the experiment. A sterile 5-inch vet premium hypodermic needle (Air-Tite, North Adams, MA, USA) was also inserted through the septum, and was affixed with a sterile 1mL Luer-Lok syringe (BD Biosciences; San Jose, CA, USA) so as to apply back-pressure. The bottles were placed in a water bath, where they were held at 30 °C for the duration of the experiment. This temperature is within the range of optimal fermentation temperatures for *K. marxianus* (30 °C to 37 °C) [22], *S. cerevisiae* (Hornindal kveik, 20 °C to 35 °C) [23] and *B. bruxellensis* and *B. claussenii* (15 °C to 37 °C) [24].

#### 2.1.2. Data Collection and Analysis

The fermentations were carried out in triplicate. During each trial period, for each fermentate, samples for density, ethanol, cell count, and organic acid and sugar profile analyses were taken via the attachment of sterile Luer-Lok syringes to the inserted needle. Samples for density were taken daily, samples for cell count were taken every other day, and samples for ethanol concentration and organic acid and sugar profile analyses were taken at the beginning and the end of the fermentations. Density was analyzed with an Anton Paar DMA 35 (Graz, Austria) densitometer. Cell count samples were serially diluted in phosphate-buffered saline solution and then replicate-plated on potato dextrose agar supplemented with chloramphenicol. Each yeast strain’s plates were incubated at 30 °C for a duration appropriate for the growth of that strain’s colonies; for each treatment, replicate plates of a dilution containing numbers of CFUs within the countable range were then counted using a Chemopharm^®^ Color QCount model 530 (Advanced Instruments, Inc., Norwood, MA, USA). Concentrations of ethanol, organic acids, and sugars were analyzed using high-performance liquid chromatography (HPLC) at the Cornell AgriTech facilities in Geneva, New York. pH and temperature were recorded every four hours by the iCinac program described above. JMP^®^ Pro 15.0.0 (SAS Institute, Cary, NC, USA) was used to calculate means and standard deviations for the three biological replicates in this experiment, and to subsequently conduct analyses of variance (ANOVAs).

### 2.2. Sensory Study

The study protocol was reviewed and approved by Cornell University Institutional Review Board (Ithaca, NY, USA) before commencement of the study. A recruitment email was sent through the Cornell Sensory Evaluation Center, and through the Cornell Food Science undergraduate and graduate listservs. Flyers were distributed in Stocking Hall, Cornell University (Ithaca, NY, USA). An online screening questionnaire, facilitated by Cornell Qualtrics (Qualtrics, Provo, UT, USA), was used to select eligible participants. Of the 83 respondents, those who were selected to participate indicated that they were above 21 years of age, not intolerant to dairy or alcohol, not pregnant, able to participate in all components of the sensory study, very comfortable with expressing opinions in a group setting, and consumed fermented alcoholic beverages at least two to three times a month. Fourteen participants, eight female and six male, aged 22 to 68, were recruited for the study; all of them provided informed consent. For each participant, the study consisted of a training session discussion followed by an individual online questionnaire. As there are typically fewer than 12 participants in a sensory study training session [25], the participants were assigned to two training sessions (*n* = 6 and *n* = 8) hosted on the same day, based on their availability. During the training sessions (described in detail below), the participants were instructed to taste five fermented YAW beverage samples, as well as nine reference samples. On the day following the training sessions, the participants were asked to taste the five test samples again while completing the individual online questionnaire (described in detail below). In adherence to the social distancing guidelines for COVID-19, this study was completed at the participants’ homes instead of at the Cornell Sensory Evaluation Center (Ithaca, NY, USA). Each participant received a $15 cash reward for their participation in this study.

#### 2.2.1. Sample Preparation

To prepare the fermentates that were to be served to individuals participating in the sensory study (described below), bottles of sterile YAW were each inoculated with one of the five yeast strains in the same ratios used in the fermentation trials, as described previously, and were fitted with an air lock. The bottles were kept in a 30 °C incubator for 7 to 22 days, depending on the yeast strain (7 days for *S. cerevisiae* (IOCBF), 12 days for *B. bruxellensis*, 17 days for *K. marxianus* and *S. cerevisiae* (Hornindal kveik), and 22 days for *B. claussenii*). Similar to the fermentation characterization trials described above, each fermentate was considered to be finished once its density was measured to be consistent for 3 consecutive days. However, with these fermentates, density samples were taken aseptically only at the beginning and near the end of the fermentation, to avoid disturbing the anoxic conditions in the bottles. The fermented beverages were then stored in a 4 °C refrigerator until the sensory study took place, 45 days after the initiation of the fermentation. The density and pH of each beverage was measured both at completion of fermentation and at two days prior to the sensory study, to confirm that no significant changes had occurred during the storage period. Yeast sediments in the fermented beverages were removed by carefully decanting the liquid portion into new containers. Reference samples were selected to represent different sensory phenomena relevant to the consumption of the prototype beverages, and included the following: 2% sucrose solution (sweetness), 0.5% sodium chloride solution (saltiness), 0.03% lactic acid solution (lactic sourness), rice vinegar diluted 1:10 (acetic sourness), 100% pineapple juice (fruity flavor), YAW (dairy flavor), non-alcoholic and non-flavored seltzer (fizziness), 0.03% caffeine solution (bitterness), and a 1:1 ratio of Fleischmann’s instant dry yeast (Ab Mauri Food Inc., St. Louis, MO, USA) and *S. cerevisiae* (IOCBF) in water suspension (1 g of mixed yeast/50 mL of water, representing yeasty flavor). The compositions and concentrations of these reference solutions were adapted from Sensory Lexicon: Unabridged Definition and Reference [26], as well as the sensory study on Mihalic cheese conducted by Aday and Karagul Yuceer (2014) [27]. The participants received all of the samples packaged in a container box the day before the study. For the training session, the participants were given 2 oz of each test and reference sample. For the individual questionnaire, they were given 5 oz of the sample fermented by *S. cerevisiae* (IOCBF) and 1 oz of each of the other four test samples. They were instructed to refrigerate the samples upon receipt, and to transfer them to room temperature one hour before the tastings. 

#### 2.2.2. Training Session

The training session was hosted over an online meeting using Zoom (Zoom Video Communications, Inc., San Jose, CA, USA). The participants were first introduced to the reference samples to achieve consensus on the definitions of basic sensory attributes that these reference samples represent. In order to encourage the participants to notice the subtle differences between the samples fermented with different yeast strains, the sample fermented with *S. cerevisiae* (IOCBF) was then introduced and used as a baseline reference. This sample was selected as the baseline reference due to the fact that *S. cerevisiae* (IOCBF) is a convenient, ready-to-use active dried yeast that is more widely used in the fermented beverage industry compared to the other strains in this study. The other four test samples were introduced subsequently. During each tasting, the participants were guided to discuss their perceptions of the sample, focusing on its evocation of particular sensations within the modalities of aroma, flavor, mouthfeel or texture, and aftertaste. At the end of the session, the group engaged in a review of the sensory descriptors generated during the discussions, to select the terms they most agreed upon. The selected terms were then further categorized and condensed under the sensory modality categories mentioned above.

#### 2.2.3. Individual Online Questionnaire

The selected sensory descriptors were then inserted into a questionnaire generated using the RedJade program (RedJade Software Solutions, LLC, Redwood City, CA, USA), and were used as metrics against which the samples were rated. In the questionnaire, participants were asked to rate the intensity of the sensory attributes of each sample on a scale of 0–10, with 0 denoting “none” and 10 denoting “extremely strong”. For the attribute “body”, 0 represented “light” and 10 represented “full”. All of the samples were labeled with a 3-digit code, and were presented to the participants in a randomized order generated by the RedJade program. To prompt the participants to compare the samples and notice their subtle differences, the sample fermented by *S. cerevisiae* (IOCBF) was presented first to all participants, and was used as a baseline reference. This reference was re-introduced before each subsequent tasting. 

A follow-up session of the study was conducted one week later, which utilized the same questionnaire to characterize the sensory profile of unfermented YAW. Seven of the original 14 participants (three female and four male) were available to participate. They were each rewarded with $5 cash for this session.

#### 2.2.4. Analyses

Responses to the questionnaire were recorded and analyzed using RedJade. Standard deviations, ANOVA, and Tukey’s honestly significant difference post-hoc test (Tukey’s HSD) were also conducted using RedJade; for the ANOVA and Tukey’s HSD analyses, the alpha value was established at 0.05.

## 3. Results and Discussion

### 3.1. Fermentation Profile

Density, pH, cell count, ethanol content, organic acid content, and sugar profile were all important markers used to track fermentation progress. The fermentates of all five strains started at a density of 1.020 ± 0.003 g/cm^3^ and a pH of 4.44 ± 0.10. The change in density of each fermentate was used as an indicator to track the progress of the fermentation. For each yeast strain investigated, the duration of the fermentation process was determined by the point during the first biological replicate at which the fermentation reached a density which was consistent for 3 consecutive days; those same durations were then artificially imposed upon the other two replicates, regardless of the trends of the densities during those replicates. According to the density measurements (Figure 1), the five yeast strains fermented at different rates, and took between 8 and 21 days to complete their respective fermentations (8 days for *S. cerevisiae* (IOCBF), 15 days for *B. bruxellensis*, 20 days for *K. marxianus* and *S. cerevisiae* (Hornindal kveik), and 21 days for *B. claussenii*). Among all of the strains, *S. cerevisiae* (IOCBF) was the fastest-acting, decreasing the density dramatically in the first 4 days of its fermentation and reaching stability by day 8. In contrast, *B. claussenii* fermented much more slowly and brought about a steady decrease in density through day 14. Interestingly, *K. marxianus* and *S. cerevisiae* (Hornindal kveik) showed highly similar trends in their effects on density, both demonstrating a moderate rate of decrease through day 8. Overall, these four strains were able to utilize almost all of the sugars available in the substrate (as substantiated by the HPLC results discussed below), decreasing the density from around 1.02 g/cm^3^ to around 1.004 g/cm^3^, which is almost the same as the density of water. *B. bruxellensis* was the only strain that stopped fermenting at a higher density (1.013 ± 0.003 g/cm^3^). It displayed a similar rate of fermentation to that of *B. claussenii* for the first 5 days, but its rate subsequently slowed significantly.

Similar trends were observed in the pH curves of the samples (Figure 2). *S. cerevisiae* (IOCBF) caused a rapid drop in pH during the first 2 days (a decline of 0.20 ± 0.03 pH units), after which the pH of its fermentate increased slightly during the remainder of the fermentation. *B. claussenii* generated the most gradual drop in pH among all investigated strains until day 6, and produced a final pH of 4.28 ± 0.14. The fermentates of *K. marxianus* and *S. cerevisiae* (Hornindal kveik) mirrored each other and underwent a moderate decrease in pH before day 5. *B. bruxellensis* caused a slightly faster drop in the pH of its fermentate than *B. claussenii* did in the first 3 days; subsequently, the pH of its fermentate remained stable from day 4 to day 8, before decreasing again until the end of the fermentation. Contrary to what one might expect to see based on the density results, *B. bruxellensis* ultimately proved capable of fermenting the substrate to a final pH (4.26 ± 0.09) similar to those reached by the other strains. Overall, the changes in pH for the fermentates of all five strains were small, and the differences between them were minor when compared to the variations seen between trials, which was largely due to the variability of the fermentation substrate YAW. For all five strains, the pH of the fermented samples stabilized before the density.

The cell counts (Figure 3) of *K. marxianus*, *B. bruxellensis*, *B. claussenii* and *S. cerevisiae* (IOCBF) were similar at the end of the fermentations, estimated between 5.26 ± 0.27 log CFU/mL and 6.02 ± 0.17 log CFU/mL. *S. cerevisiae* (Hornindal kveik) had a significantly lower final cell count than did all of the other strains (4.17 ± 0.33 log CFU/mL, *p* < 0.01). During its fermentation, *S. cerevisiae* (IOCBF) underwent an increase of roughly 1 log CFU/mL in the first two days, but then steadily decreased over the following six days to 5.23 ± 0.27 log CFU/mL. The cell count of *B. claussenii* also increased slightly in the first 4 days, before decreasing at a nearly constant rate between day 4 and day 10, reaching a final count of 5.61 ± 0.66 log CFU/mL on day 20. *B. bruxellensis*, *K. marxianus* and *S. cerevisiae* (Hornindal kveik) all decreased steadily in cell count by approximately 2 log CFU/mL before day 6. While *K. marxianus* showed no statistically significant change in cell count after day 6, *S. cerevisiae* (Hornindal kveik) increased by 0.66 log CFU/mL (*p* = 0.04) and *B. bruxellensis* increased back to its original cell count during the rest of its fermentation. Throughout the course of this experiment, *K. marxianus* and *B. claussenii* dropped by about 1 log CFU/mL (*p* < 0.01 and *p* = 0.04 respectively), whereas *S. cerevisiae* (Hornindal kveik) dropped by 1.63 log CFU/mL (*p* = 0.01). The other yeast strains experienced no significant changes in cell count. The estimated cell numbers may have been lower than the true values, as we refrained from homogenizing the contents of the fermentation bottles prior to aspirating samples, in an attempt to avoid disturbing their anoxic conditions.

As shown in Table 1, the substrate’s initial concentrations of lactose, galactose, and glucose varied, depending on the inoculating strain and the presence or absence of supplemental lactase. With regard to the fermentates that received supplemental lactase, the lactose was hydrolyzed into 19.19 ± 3.22 g galactose and 14.65 ± 1.79 g glucose per liter. After fermentation, *B. bruxellensis* was the only strain that had left behind residual galactose (20.28 ± 4.91 g/L), while all other strains were able to utilize all of the available sugars. *B. bruxellensis* also produced less ethanol (1.07 ± 0.04%) compared to all other strains (2.33 ± 0.38%); this fact can be explained by *B. bruxellensis*’s inability to ferment galactose. This finding aligns with the density results, wherein the *B. bruxellensis* fermentate failed to reach the same final density as those of the other strains. There were no significant differences between the initial and final concentrations of citric acid, lactic acid, and acetic acid for any of the fermentates.

Large variations in density, pH, and HPLC results were observed between the three biological replicates, as indicated by their standard deviations. This is due to the fact that a separate batch of YAW was used in each biological replicate of this experiment. Since the YAW was a by-product of dairy processing, its chemical composition could not be controlled; this caused variations in the initial conditions of the fermentation trials.

### 3.2. Sensory Profile

Twenty-six sensory descriptors were generated during the training session discussion (as listed in Table 2). The sensory profiles of the five fermented YAW samples and the unfermented YAW are plotted on radar charts for purposes of visualization (Figure 4 and Figure 5). Prior to undergoing fermentation, YAW was perceived as similar to yogurt, with the aroma and flavor of lactic acid or dairy being the dominant sensory attributes (Figure 4). According to Gallardo-Escamilla and colleagues (2005) [28], the odors of rancidity and yogurt and the flavors of acidity, rancidity, and fruit were the main sensory attributes of whey that had been produced via fermentation of milk with yogurt cultures. The sensory profile of YAW has been demonstrated to vary depending on the starter cultures and processing methods used. 

During the fermentations outlined in the current research, each of the yeast strains caused the development of distinct sensory characteristics, and dramatically transformed the sensory profile of the YAW. Among the five sensory modalities evaluated by the participants, the main differences perceived between the samples were concentrated in the areas of aroma and appearance. In particular, significant differences were observed between the five samples with regard to the following descriptors: fruity apple aroma (*p* < 0.01); the aromas of dairy and of acidity, specifically lactic acid (*p* = 0.01), the aromas of bakeries and of yeast (*p* = 0.04), the odors of chemicals such as petroleum, gasoline, and solvents (*p* = 0.01); and cloudiness (*p* < 0.01). Borderline significant differences were observed with respect to the descriptors: odor of poorly aged or rancid cheese or milk (*p* = 0.05), and body (*p* = 0.05). YAW fermented by *S. cerevisiae* (IOCBF) was characterized as having aromas of fruity apple, bakeries, yeast, dairy, and acidity (specifically lactic acid). The perceived fruity apple aroma is consistent with the fact that this particular yeast strain is known to produce high levels of ethyl and acetate esters, which are generally associated with fresh fruit aromas, strawberry, confectionery, candy, and citrus aromas [29]. Intensely fruity wines fermented by certain strains of *S. cerevisiae* are also found to contain significant amounts of ethyl esters such as ethyl octanoate (associated with a sweet aroma), ethyl hexanoate (a green apple aroma), and ethyl butyrate (the aroma of an acidic fruit) [30]. It is likely that these volatiles were present in the YAW sample fermented by *S. cerevisiae* (IOCBF). Some strains of *S. cerevisiae* have been reported to produce sulfur compounds that can bind to acetaldehyde, a volatile that plays an important role in the flavor of yogurt [31,32,33]. The IOCBF strain used in this study, in contrast, is marketed as producing very low levels of sulfur compounds such as sulfur dioxide and hydrogen sulfide [29]. Differing retention rates of acetaldehyde may provide an explanation for the perceived difference in the aromas of dairy and lactic acid in the sample fermented by *S. cerevisiae* (IOCBF), as compared to those fermented by *B. bruxellensis* and *B. claussenii*. It is possible that the *Brettanomyces* strains were able to suppress this aroma by producing sulfur compounds that could inhibit the volatilization of the YAW’s endogenous acetaldehyde.

When a comparison was made between the aromas produced by the two *Brettanomyces* strains, the sample fermented by *B. claussenii* received a higher rating for fruity apple aroma; with regard to the sample fermented by *B. bruxellensis*, on the other hand, participants noted a heightened perception of the aromas of poorly aged or rancid cheese, and those of chemicals (petroleum, gasoline, or solvent), although these differences between the two fermentates were not statistically significant. Some research has been conducted to characterize the sensory profiles produced by *Brettanomyces* fermentations; however, *B. bruxellensis* has been more thoroughly researched than *B. claussenii*. The flavors commonly associated with *Brettanomyces* are “rubber”, “burnt plastic”, “medicinal”, “leather”, and “barnyard” [34]. These terms neatly match the participants’ perceptions of the sample fermented by *B. bruxellensis*. It is interesting to note that even though there was residual sugar in the *B. bruxellensis*-fermented sample, as indicated by both its final density and the HPLC results, the participants did not perceive any sweetness in this sample during the training session. Studies have shown that strains of *B. bruxellensis* vary in their utilization of galactose [35,36]; as evidenced by the HPLC results (Table 1), the *B. bruxellensis* strain used in this study was able to ferment the glucose, but not the galactose, generated by hydrolysis of the lactose in the YAW. It is understandable that the residual galactose did not contribute significantly to any perceived sweetness of the fermented beverage, as galactose has a relatively low sweetness index (32% of the perceived sweetness of sucrose [37]). With respect to *B. claussenii*, more research must be conducted to understand the sensory impacts of this strain on other fermentation substrates, such as grape must, in order to see if this strain produces fruity aromas consistently.

Similar to *B. bruxellensis*, *K. marxianus* also produced an aroma that the participants associated with chemicals. Other researchers have previously demonstrated that differences in fermentation medium composition have an influence on the aroma compounds produced by *K. marxianus* [35,36]. When glucose was the main ingredient in a substrate supplemented with sources of nitrogen and phosphate, *K. marxianus* was found to produce high levels of phenylethyl alcohol, which contributes to a floral odor [38]. The production of acetate esters is also particularly stimulated by a high concentration of glucose in the substrate, and these esters are primarily responsible for the fruity flavors in fermented beverages [39]. The low intensity of fruity notes perceived in the sample fermented by *K. marxianus* in this sensory study may be due to the lack of glucose in YAW, since no lactase was added to the substrate for this strain. It is unclear what might account for the chemical aroma of this beverage sample, but a contribution could be a combination of various compounds produced by the yeast’s metabolism. A chemical analysis of the fermented beverage needs to be conducted, to gain insights into the underlying mechanisms of this phenomenon.

Although there may have been slight differences between the five beverages in terms of the amount of yeast sediment that was unintentionally retained in the tasting samples, the percentage of solids that remained suspended in the liquid should have been similar in all five samples. Following fermentation but before decanting, they had similar cell counts, as indicated by the end points of the curves in Figure 3. The exception to this was the sample fermented by *S. cerevisiae* (Hornindal kveik), which had a cell count that was 1 log CFU/mL lower than the rest. Despite all steps taken to standardize the percent solids, *S. cerevisiae* (IOCBF) received a rating of 4.74 ± 2.62 for cloudiness, which was significantly higher than the ratings of all other strains (*p* < 0.01). In addition, *S. cerevisiae* (IOCBF) received the highest ratings for body, the mean of which was 3.72 ± 2 (*p* = 0.05). The reasons for these sensory attributes are unclear, and further research is required to study the effects of yeast cell suspension and lysis on the appearance and mouthfeel of these fermented beverages. The low cell count and relatively light body of the *S. cerevisiae* (Hornindal kveik)-fermented sample could both be related to the fact that this strain has a tendency to flocculate and, therefore, a larger portion of the yeast cells likely sedimented and were subsequently separated from the liquid [40]. 

## 4. Conclusions

Findings from this study provide a foundational understanding of the behaviors and resulting sensory characteristics of five yeast strains in YAW. *S. cerevisiae* (IOCBF) was the fastest-fermenting strain, and its fermentate was perceived as having aromas associated with fruity apple, yeast, bakeries, dairy, and acidity, particularly lactic acid. *B. clausenii* took the most time to complete its fermentation, but was also able to produce a fruity apple aroma. *B. bruxellensis* was the only strain that did not utilize all the sugars in the substrate, and its fermentation resulted in an aroma of poorly aged or rancid cheese or milk, and of chemicals. *K. marxianus* and *S. cerevisiae* (Hornindal kveik) displayed similar fermentation profiles, but the former produced a characteristic chemical aroma while the latter produced a milder sensory profile compared to those of the other strains. The *Brettanomyces* strains were able to significantly reduce the strong aromas of lactic acid and dairy in YAW. This study presents prototypes of novel yeast-fermented beverages as a potentially sustainable solution to the problem of overproduction of the dairy processing by-product, YAW. With the knowledge gained from this study, future products can be developed based on the considerations of fermentation rate as well as nutritional and sensory profiles of the product. A sensory acceptability test and a chemical analysis are required to better understand consumers’ responses to these prototypes, and to optimize the sensory profiles accordingly.

## Figures and Tables

**Figure 1 foods-10-01204-f001:**
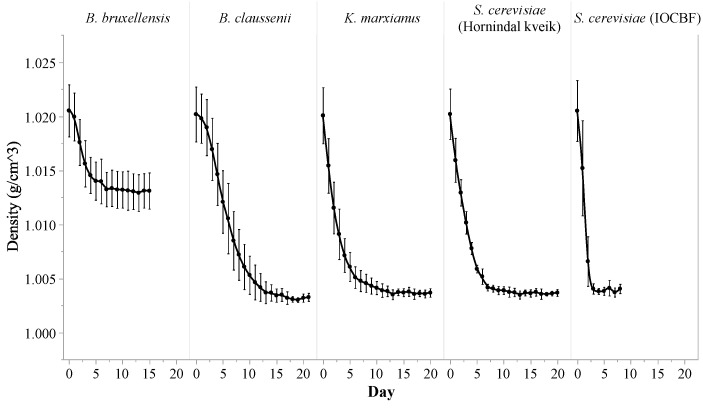
Density profiles of the fermentates of *Kluyveromyces marxianus*, *Saccharomyces cerevisiae* (Hornindal kveik), *Brettanomyces bruxellensis*, *Brettanomyces claussenii* and *Saccharomyces cerevisiae* (IOCBF) during their respective fermentations of yogurt acid whey. Values represent means ± standard deviations.

**Figure 2 foods-10-01204-f002:**
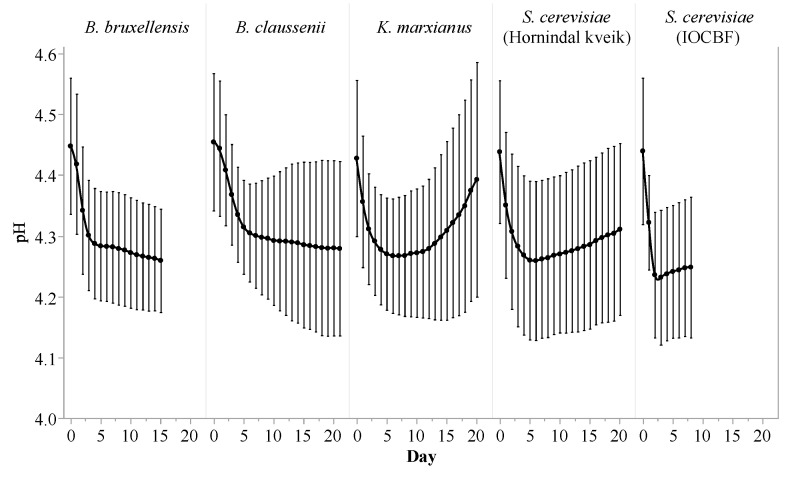
pH profiles of the fermentates of *Kluyveromyces marxianus*, *Saccharomyces cerevisiae* (Hornindal kveik), *Brettanomyces bruxellensis*, *Brettanomyces claussenii* and *Saccharomyces cerevisiae* (IOCBF) during their respective fermentations of yogurt acid whey. Values represent means ± standard deviations.

**Figure 3 foods-10-01204-f003:**
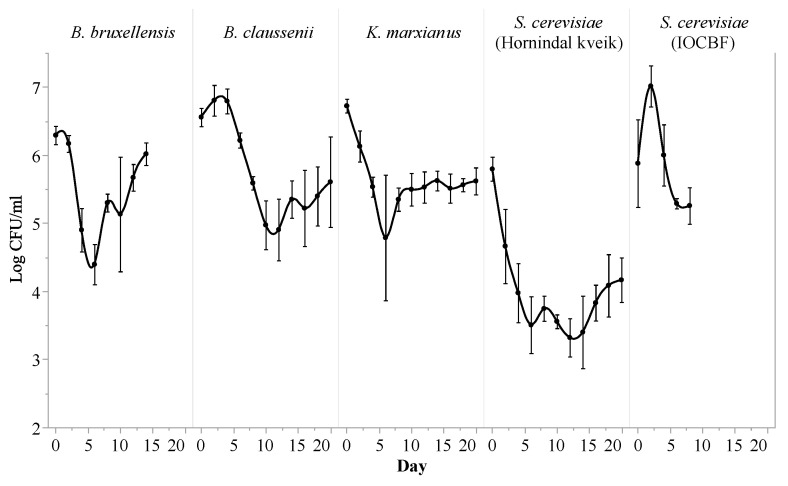
Cell count profiles of *Kluyveromyces marxianus*, *Saccharomyces cerevisiae* (Hornindal kveik), *Brettanomyces bruxellensis*, *Brettanomyces claussenii* and *Saccharomyces cerevisiae* (IOCBF) during their respective fermentations of yogurt acid whey. Values represent means ± standard deviations. The displayed cell count of *S. cerevisiae* (Hornindal kveik) on day 12 represents the mean of only two biological replicates in this experiment, with standard deviation indicated by error bars.

**Figure 4 foods-10-01204-f004:**
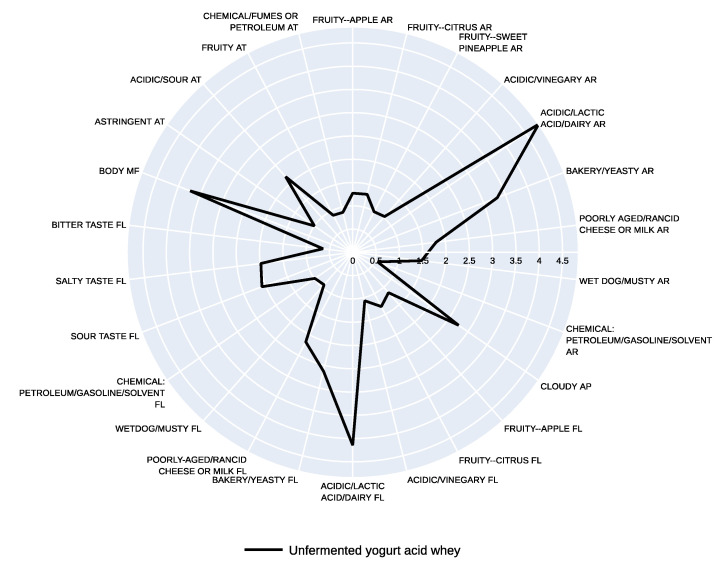
Sensory profile of unfermented yogurt acid whey. Values represent means of sensory ratings. Abbreviations: AR: aroma, AP: appearance, FL: flavor, MF: mouthfeel, AT: aftertaste.

**Figure 5 foods-10-01204-f005:**
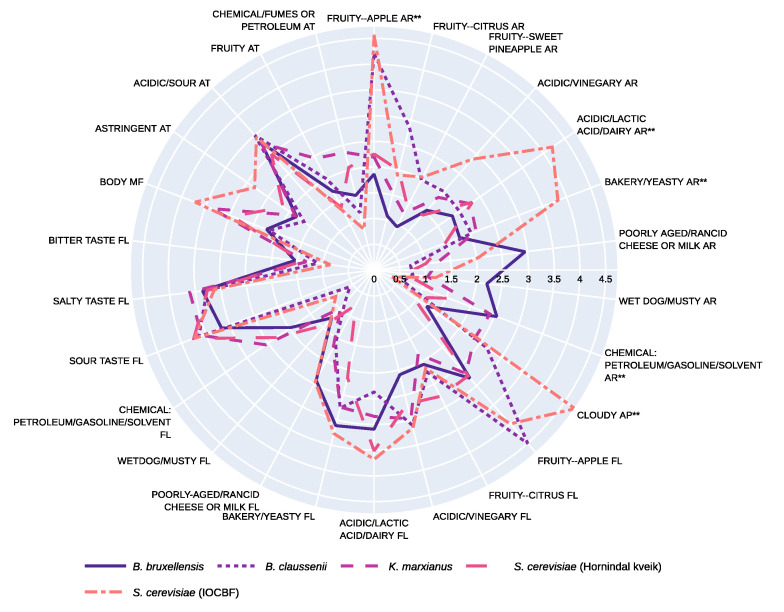
Sensory profiles of yogurt acid whey beverages fermented by *Kluyveromyces marxianus*, *Saccharomyces cerevisiae* (Hornindal kveik), *Brettanomyces bruxellensis*, *Brettanomyces claussenii*, and *Saccharomyces cerevisiae* (IOCBF). Values represent means of sensory ratings. Abbreviations: AR: aroma, AP: appearance, FL: flavor, MF: mouthfeel, AT: aftertaste. “**” indicates statistically significant difference (*p* < 0.05) between at least two of the five samples.

**Table 1 foods-10-01204-t001:** Concentrations of organic acids, ethanol, and sugars (means and standard deviations) in yogurt acid whey at beginning and end of fermentations by *Kluyveromyces marxianus*, *Saccharomyces cerevisiae* (Hornindal kveik), *Brettanomyces bruxellensis*, *Brettanomyces claussenii* and *Saccharomyces cerevisiae* (IOCBF).

	Citric Acid (g/L)	Lactic Acid (g/L)	Acetic Acid (g/L)	Ethanol (%)	Lactose (g/L)	Galactose (g/L)	Glucose (g/L)
Beginning of fermentation							
*K. marxianus*	2.00 ± 0.28	6.60 ± 0.55	1.18 ± 0.34	0.15 ± 0.08 **	32.85 ± 6.65 **	3.13 ± 0.13 **	1.06 ± 1.83
*S.cerevisiae* (Hornindal kveik)	2.00 ± 0.27	6.48 ± 0.42	1.04 ± 0.37	0.14 ± 0.08 **	4.13 ± 3.33	19.12 ± 3.43 **	14.31 ± 1.55 **
*B. bruxellensis*	2.02 ± 0.27	6.52 ± 0.37	1.07 ± 0.26	0.12 ± 0.10 **	4.01 ± 3.11	19.06 ± 2.84	14.73 ± 1.67 **
*B. claussenii*	2.01 ± 0.27	6.58 ± 0.48	1.14 ± 0.42	0.12 ± 0.10 **	33.26 ± 6.51 **	4.22 ± 1.61 **	nd
*S.cerevisiae* (IOCBF)	2.02 ± 0.28	6.58 ± 0.55	1.18 ± 0.65	0.13 ± 0.11 **	3.70 ± 3.12	19.38 ± 3.35 **	14.91 ± 2.09 **
End of fermentation							
*K. marxianus*	1.95 ± 0.25	5.97 ± 0.84	1.78 ± 0.60	2.33 ± 0.42 **	nd **	nd **	nd
*S.cerevisiae* (Hornindal kveik)	1.94 ± 0.21	6.01 ± 0.41	0.53 ± 0.24	2.28 ± 0.30 **	nd	nd **	nd **
*B. bruxellensis*	1.99 ± 0.31	6.30 ± 0.37	1.26 ± 0.12	1.07 ± 0.04 **	nd	20.28 ± 4.91	nd **
*B. claussenii*	1.99 ± 0.40	6.47 ± 1.09	1.16 ± 0.13	2.41 ± 0.37 **	nd **	nd **	nd
*S.cerevisiae* (IOCBF)	2.01 ± 0.28	6.11 ± 0.46	0.91 ± 0.37	2.30 ± 0.40 **	nd	nd **	nd **

nd = not detected. “**” indicates statistically significant difference (*p* < 0.05) observed when comparing concentrations at beginning and end of fermentation for each yeast strain.

**Table 2 foods-10-01204-t002:** Sensory ratings (means and standard deviations) and results of analysis of variance (ANOVA) for unfermented yogurt acid whey (YAW) and YAW beverages fermented by *Kluyveromyces marxianus*, *Saccharomyces cerevisiae* (Hornindal kveik), *Brettanomyces bruxellensis*, *Brettanomyces claussenii*, and *Saccharomyces cerevisiae* (IOCBF).

	*K. marxianus*	*S. cerevisiae* (Hornindal Kveik)	*B. bruxellensis*	*B. claussenii*	*S. cerevisiae* (IOCBF)	Unfermented YAW
Aroma						
Fruity--apple	2.19 ± 1.89 ^ab^	2.26 ± 1.85 ^ab^	1.86 ± 1.96 ^b^	4.20 ± 2.53 ^ab^	4.57 ± 2.62 ^a^	1.27 ± 1.00
Fruity--citrus	1.54 ± 0.96 ^a^	2.00 ± 1.90 ^a^	1.08 ± 1.24 ^a^	2.86 ± 2.13 ^a^	1.90 ± 1.83 ^a^	1.29 ± 1.50
Fruity--sweet pineapple	1.24 ± 1.15 ^a^	1.39 ± 1.82 ^a^	0.95 ± 1.58 ^a^	1.98 ± 1.33 ^a^	2.04 ± 2.37 ^a^	0.99 ± 1.26
Acidic, vinegary	1.89 ± 1.31 ^a^	1.43 ± 1.27 ^a^	1.55 ± 1.31 ^a^	2.04 ± 2.00 ^a^	2.88 ± 2.47 ^a^	1.03 ± 0.90
Acidic, lactic acid, dairy	2.27 ± 1.44 ^ab^	2.31 ± 2.02 ^ab^	1.85 ± 1.85 ^b^	2.01 ± 1.38 ^b^	4.21 ± 2.25 ^a^	4.84 ± 3.24
Bakery, yeasty	2.12 ± 2.26 ^ab^	1.48 ± 1.48 ^b^	1.78 ± 1.45 ^ab^	2.02 ± 2.18 ^ab^	3.82 ± 2.59 ^a^	3.31 ± 2.73
Poorly aged, Rancid cheese or milk	1.24 ± 1.39 ^a^	1.01 ± 2.16 ^a^	2.96 ± 2.65 ^a^	0.71 ± 1.07 ^a^	2.05 ± 2.68 ^a^	1.80 ± 3.04
Wet dog, musty	1.05 ± 1.21 ^a^	0.61 ± 0.90 ^a^	2.21 ± 2.74 ^a^	0.76 ± 1.04 ^a^	1.21± 2.40 ^a^	1.49 ± 3.09
Chemical: petroleum, gasoline, solvent	2.41 ± 2.55 ^ab^	1.55 ± 1.58 ^ab^	2.55 ± 2.73 ^a^	0.60 ± 0.85 ^ab^	0.38 ± 0.57 ^b^	0.57 ± 1.02
Appearance						
Cloudy	2.36 ± 1.08 ^b^	0.93 ± 0.71 ^b^	1.24 ± 1.00 ^b^	2.66 ± 2.50 ^b^	4.74 ± 2.62 ^a^	2.76 ± 1.54
Flavor						
Fruity--apple	2.69 ± 2.15 ^a^	2.75 ± 2.10 ^a^	2.81 ± 2.01 ^a^	4.50 ± 1.94 ^a^	3.99 ± 2.48 ^a^	1.16 ± 1.03
Fruity--citrus	1.84 ± 1.81 ^a^	2.77 ± 2.46 ^a^	2.07 ± 1.88 ^a^	2.25 ± 2.14 ^a^	2.16 ± 2.00 ^a^	1.31± 1.38
Acidic, vinegary	2.98 ± 1.71 ^a^	2.68 ± 1.80 ^a^	2.09 ± 2.10 ^a^	3.11 ± 2.97 ^a^	3.16 ± 2.50 ^a^	1.07 ± 0.99
Acidic, lactic acid, dairy	2.84 ± 1.64 ^a^	3.51 ± 1.52 ^a^	3.09 ± 1.89 ^a^	2.37 ± 1.30 ^a^	3.68 ± 2.23 ^a^	4.14 ± 2.36
Bakery, yeasty	2.74 ± 2.05 ^a^	2.16 ± 1.45 ^a^	3.11 ± 2.32 ^a^	2.76 ± 2.00 ^a^	3.27 ± 2.03 ^a^	2.63 ± 1.32
Poorly aged, rancid cheese or milk	1.71 ± 1.79 ^a^	0.77 ± 0.79 ^a^	2.42 ± 2.18 ^a^	1.61 ± 1.95 ^a^	2.46 ± 2.75 ^a^	2.17 ± 1.24
Wet dog, musty	0.99 ± 1.18 ^a^	1.43 ± 2.33 ^a^	1.24 ± 2.01 ^a^	0.86 ± 1.35 ^a^	1.26 ± 1.98 ^a^	0.93 ± 1.35
Chemical: petroleum, gasoline, solvent	2.56 ± 2.88 ^a^	2.31± 2.45 ^a^	1.96 ± 2.45 ^a^	0.57 ± 0.81 ^a^	0.91 ± 1.79 ^a^	0.99 ± 2.05
Sour taste	3.59 ± 1.85 ^a^	3.75 ± 1.52 ^a^	3.17 ± 2.08 ^a^	3.64 ± 1.45 ^a^	3.73 ± 2.45 ^a^	2.09 ± 1.64
Salty taste	3.61 ± 2.07 ^a^	3.26 ± 1.98 ^a^	3.36 ± 2.38 ^a^	3.27 ± 1.79 ^a^	3.13 ± 2.23 ^a^	1.99 ± 1.32
Bitter taste	1.33 ± 1.75 ^a^	1.41± 1.54 ^a^	1.55 ± 1.81 ^a^	1.11± 1.26 ^a^	0.86 ± 1.04 ^a^	0.64 ± 1.01
Mouthfeel						
Body	3.37 ± 1.68 ^a^	2.95 ± 1.46 ^a^	2.24 ± 1.20 ^a^	2.18 ± 1.15 ^a^	3.72 ± 2.00 ^a^	3.73 ± 2.30
Aftertaste						
Astringent	1.86 ± 1.59 ^a^	2.15 ± 2.02 ^a^	1.83 ± 2.09 ^a^	1.66 ± 1.46 ^a^	2.82 ± 2.49 ^a^	1.00 ± 0.81
Acidic, sour	3.30 ± 1.39 ^a^	3.01 ± 0.93 ^a^	3.31 ± 2.16 ^a^	3.49 ± 2.05 ^a^	3.45 ± 1.93 ^a^	2.17 ± 1.41
Fruity	2.45 ± 2.26 ^a^	1.38 ± 1.36 ^a^	1.73 ± 1.84 ^a^	2.01 ± 1.38 ^a^	1.29 ± 1.51 ^a^	0.90 ± 0.72
Chemical, fumes or petroleum	2.36 ± 2.75 ^a^	2.06 ± 2.32 ^a^	1.49 ± 1.87 ^a^	1.15 ± 1.19 ^a^	0.81 ± 1.53 ^a^	0.89 ± 1.71

Note: ratings for unfermented YAW were not included in ANOVA and Tukey’s honestly significant difference (HSD), due to smaller sample size (*n* = 7). ^a,b^ After Tukey’s HSD, means within a row not sharing the same superscript were significantly different (*p* < 0.05).

## Data Availability

The data presented in this study are available on request from the corresponding author.

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
