# Peer review of "Characterization of the Fermentation and Sensory Profiles of Novel Yeast-Fermented Acid Whey Beverages"

_foods, 2021, doi:10.3390/foods10061204_

Round 1

Reviewer 1 Report

The authors discussed a very important and interesting topic of using an important by-product of the dairy industry - acid whey - as a raw material for the production of yeast-fermented beverages, presenting valuable and reliable results. Nevertheless, some aspects should be addressed and need improvement.

  1. please use spaces between values and units
  2. line 54 - beverages instead of beverage
  3. autoclaving uses very high temperature and pressure which, although it kills microorganisms, causes significant changes in the raw material, which can certainly affect the further fermentation process. Have the authors considered pasteurization of raw material as an alternative method of raw material processing? After autoclaving, was microbiological control cultures performed to demonstrate sterility? 
  4. Line 151, please use two decimal places for microbial concentration values.
  5. Table 1 - statistical differences between samples before and after fermentation should be shown.
  6. Research on the health-promoting potential of the developed products should be carried out, for example, of antioxidant activity. 

Reviewer 2 Report

This MS is on the whole clearly and logically written.  The study appears to be well designed and the conclusions valid, in particular the limitations of the study such as the lack of volatile analysis and assessment of liking.

My major concerns of the study are:

  1. I found it unclear how many trials and replicates were carried out. The authors mention variability in raw material as contributing to variations in pH and other fermentation variables, however, they also state on L 140 that the YAW was obtained, autoclaved and split into 5 batches – so where is the variability?  However, if it was only split into 5 1 L bottles – where did the biological replication come from??  or is this technical replication ?  Were batches of YAW obtained on three occasions?  In any case I found this to be quite unclear.
  2. Details of what is plotted in the figures needs to be explained in the Figure legends. – values represent mean of XXX + SD ? SE ?.  Also, I believe Figs 1 to 3 are not showing “Changes” (differences from a point) – rather they show actual values of pH, density etc over time.
  3. When discussing the pH overtime and to a lesser extend cell numbers, the authors state on a number occasions that values are not significantly different but then discuss increases and decreases.     L 345.   There is nothing wrong with stating that there was no statistically significant change in pH over the time of the experiment or from day X onwards.   This section could be shortened and simplified.
  4. L 356. I believe that cell “numbers” were “estimated”  - not “concentrations” “measured” – using the concept of numbers – it is then possible to simply and more correctly state that “the estimated number of cells – increased or decreased or showed no significant difference!!
  5. Table 2. This could be put into supplementary material or removed as its content  is shown in Figs 4 and 5 and discussed in the text.
  6. Figures 4 and 5 the font for the descriptors could be improved to make them easier to read.
  7. Why were the ferments carried out at 30oC ? This is quite high compared to temps used in brewing and wine making.  Such temperatures are known to stress yeast cells and increase the concentration of esters produced.

Minor

  1. L 46 – 2012 [4] seems a very old reference – given earlier the number has increased dramatically over the last two decades ??
  2. L 49 – water “streams” or “ways”?
  3. L 60 – New Paragraph – it took me a while to figure out why this fact was thrown in here - no mention in the title ??  and tucked away in  the abstract (L31).  Maybe switch order of this and the following paragraph?  In any event I found this small section to be a little disjointed.

Reviewer 3 Report

The authors describe the fermentation of acid-whey by using five yeast strains for the beverages. The results were interesting, however, it was not clear how much the acid-whey beverages are different from conventional whey beverages. Are their tastes close? Could the authors show the profiles and scores of conventional whey beverages in tables 1 and 2? Or describe about it?

Round 2

Reviewer 1 Report

I reanalyzed the document and the answers given by the authors, and consider that the document was clearly improved, thus it can be accepted.